# Studies on the β → α Phase Transition Kinetics of Ti–3.5Al–5Mo–4V Alloy under Isothermal Conditions by X-ray Diffraction

**Panpan Ge** [1,2]**, Song Xiang** [1,2,*] 📱**, Yuanbiao Tan** [1,2] **and Xuanming Ji** [1,2]

[1] College of Materials and Metallurgy, Guizhou University, Guiyang 550025, China; panpange1995@126.com (P.G.); ybtan1@gzu.edu.cn (Y.T.); xmji@gzu.edu.cn (X.J.)

[2] Key Laboratory for Mechanical Behavior and Microstructure of Materials of Guizhou Province, Guizhou University, Guiyang 550025, China

\* Correspondence: sxiang@gzu.edu.cn; Tel.: +86-189-8515-1196

**Abstract:** The β → α phase transition kinetics of the Ti–3.5Al–5Mo–4V alloy with two different grain sizes was investigated at the isothermal temperature of 500 °C. A method to estimate the function of the precipitate fraction of the α phase with different aging times was developed based on X-ray diffraction analysis. The value of the α precipitate fraction increased sharply at first, then increased slowly with the aging time, and finally reached equilibrium. The value of the α precipitate fraction was higher in the alloy aged for the same time at a higher solution temperature, while the size of the α precipitate was smaller at a higher solution temperature. The β → α phase transition kinetics under isothermal conditions were modeled in the theoretical frame of the Johnson–Mehl–Avrami–Kolmogorov (JMAK) theory. The kinetic parameters of JMAK deduced different transformation mechanisms. The mechanism of the phase transition in the first stage was dominated by mixed transformation mechanisms (homogeneously nucleated and acicular-grown α structure, and grain boundary-nucleated and grown α precipitate), while the second stage was the growth of the fine α precipitate, which was controlled by slow diffusion. As the aging time increased, the hardness of the Ti–3.5Al–5Mo–4V alloy increased sharply. After the hardness of the alloy reached a plateau, it began to decline. The hardness of the alloy was always higher at a higher solution temperature.

**Keywords:** Ti–3.5Al–5Mo–4V alloy; aging treatment; X-ray diffraction; phase transition; kinetics; age hardening

## 1. Introduction

Metastable β-titanium alloys provide an advantageous balance of mechanical properties compared with other titanium alloys [1]. They are extensively utilized in the aircraft, automotive, and even military industries as a result of their high strength-to-density ratio, superior corrosion resistance, excellent hot and cold formability, considerable hardening ability, and high fatigue and crack propagation resistance [2–5]. Unfortunately, the molybdenum equivalent of the majority of commercial metastable β-titanium alloys is relatively high, resulting in increased alloy costs, smelting difficulty, inhomogeneous precipitation, and a slow aging response [6].

A new titanium alloy, namely, Ti–3.5Al–5Mo–4V, was designed according to the near critical molybdenum equivalent and multi-component strengthening principle [3]. The nominal molybdenum equivalence in this alloy is only 10.3, and the total content of the β stabilizing elements is about 12%; these values are almost the lowest in the metastable β-titanium alloys. The new alloy, which is

different from traditional metastable β-titanium alloys, exhibits a fast aging response during aging treatment due to its relatively lean β stabilizer content [6,7]. For example, the Ti-55521 alloy, which has a molybdenum equivalent of 12.05 has a negligible content variation between the α nuclei and β phase in the early aging stage, indicating a slow aging response of the alloy [8].

The Ti–3.5Al–5Mo–4V alloy contains a metastable β phase after quenching from above the β transus temperature to room temperature due to the existence of sufficient total β stabilizer elements, such as Cr, Mo, V, and Fe [9]. Following the solution treatment, aging treatment results in the precipitation of α dispersed in the β matrix. In fact, the grain size, morphology, distribution, and value of the α precipitate fraction have decisive effects on the increased strength in the alloy, as the fine α phase is a hard and un-deformable particle which precipitates in the β matrix and provides more α/β interfaces to hinder the dislocation movement [10–12]. In general, compared with solution treatment, aging treatment is a more important method to enhance the mechanical performance of titanium alloys, which strongly depends on the heat treatment temperature and soaking time [13]. Therefore, it is of great industrial significance and scientific interest to improve our understanding of the response of titanium alloys to different heat treatments, such as where the α phase can precipitate, and of the microstructural evolution during the β → α phase transition [14]. A comprehensive understanding of the β → α phase transition kinetics can help to better determine the critical temperatures that dominate the distribution of the α phase, understand the mechanisms of precipitation, and improve or optimize the mechanical properties (strength, toughness, hardness, etc.) of titanium alloys.

It is noteworthy that the mechanical properties of titanium alloys are usually functions of the fraction of the α precipitate. It is particularly significant to obtain a method to quantitatively determine the degree of the precipitated α phase that can also simulate the precipitation reaction [14]. Malinov et al. [15,16] used Thermo-Calc software to quantitatively calculate the value of the α precipitate fraction according to the Ti database for the titanium alloys. Bruneseaux et al. [17] applied electrical resistivity measurement for this calculation for the Ti17 titanium alloy. A similar study was carried out on the β-CEZ alloy [18]. However, it is difficult to accurately detect the volume fraction of α precipitation using electrical resistivity measurements when the precipitate fraction is relatively low. Kherrouba et al. [19] carried out such calculations with differential scanning calorimetry (DSC) for the titanium alloy. A similar study was carried out on the Ti–10V–2Fe–3Al alloy [20]. Unfortunately, the deviation of the DSC results was slightly higher. It is worth noting that X-ray diffraction (XRD) provides an accurate and reliable examination method for quantitative calculations. Even though some researchers used X-ray diffraction based techniques to carry out quantitative analyses of the content of precipitates in steels, such as 304 stainless steel [21] and maraging steel [22], the direct comparison methodology using X–ray diffraction is rarely employed to quantitatively determine the precipitation fraction of the α phase in titanium alloys. Undoubtedly, this plays a pivotal role in the study of the phase transition kinetics.

Many researchers studied the phase transition kinetics of titanium alloys [23–27]. Behera et al. [23] studied the influence of the cooling rate on the β → α phase transition mechanism of Ti–5Ta–1.8Nb alloy under continuous cooling conditions. They found that increasing the cooling rate suppressed the β → α transformation temperature domain, leading to a composition variation of the α and β phases. Naveen et al. [24] studied the β → α + β phase transformation kinetics of Ti–15V–3Cr–3Al–3Sn alloy under isothermal conditions, with results showing that the mechanism of the phase transition was the growth of the fine α precipitate, which was controlled by slow diffusion. Appolaire et al. [25] presented a model to describe the evolution of high temperature microstructures in the β-CEZ alloy under isothermal treatments. Settefrati et al. [26] studied the β phase transformation kinetics of the Ti-5553 alloy under a low aging temperature, indicating the transformation with poor solute partitioning. Behera et al. [27] used differential scanning calorimetry to study the effect of cooling and heating rates on the α → β phase transformation kinetics of the Ti–4.4 mass% Ta–1.9 mass% Nb alloy. They found that the transformation starting ($T_s$), peak ($T_p$), and finishing ($T_f$) temperatures showed systematic variation with the cooling and heating rates, thereby leading to a kinetic widening of the phase region.

These factors prompted us to focus on characterizing the phase transition kinetics of a newly developed Ti–3.5Al–5Mo–4V metastable β-titanium alloy under isothermal conditions. In this study, the Ti–3.5Al–5Mo–4V alloy was solution-treated at different temperatures to produce a β phase with two different grain sizes. The β → α phase transition kinetics under isothermal conditions of the Ti–3.5Al–5Mo–4V alloy with two different grain sizes were studied using X-ray diffraction, and a model was built using the Johnson–Mehl–Avrami–Kolmogorov (JMAK) analysis framework. This analytical method has been widely used in solid-state phase transition kinetics studies [28–32]. Hardness testing was carried out to study the evolution of the mechanical properties in the process of β → α phase transformation.

## 2. Materials and Methods

The Ti–3.5Al–5Mo–4V alloy was produced via vacuum arc remelting for this study [6]. The chemical composition of the alloy investigated is given in Table 1. By way of producing the β phase with two different grain sizes, solution treatment of the alloy was in the β phase field (which was held at 850 and 1050 °C for 1 h), followed by quenching in water (WQ). Following this, the alloy was cut into several pieces for aging treatment. These pieces were isothermally aged at 500 °C for different times ranging from 2 min to 48 h in a salt bath. All pieces were air-cooled (AC) to room temperature after aging treatment. The schematic of the heat treatment routine is shown in Figure 1.

**Table 1.** The chemical composition of the Ti–3.5Al–5Mo–4V alloy.

| Element | Al | Mo | V | Cr | Sn | Zr | Fe | C | N | O | H | Ti |
|---|---|---|---|---|---|---|---|---|---|---|---|---|
| Mass percent/% | 3.62 | 4.83 | 3.86 | 2.09 | 1.98 | 2.02 | 1.01 | 0.032 | 0.019 | 0.007 | 0.002 | Bal. |

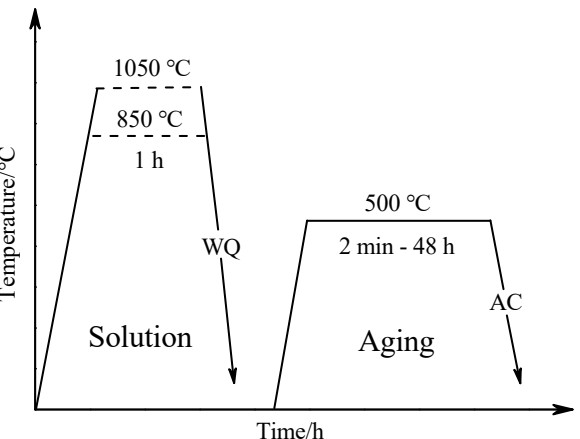

**Figure 1.** The schematic illustration of heat treatment routine of the Ti–3.5Al–5Mo–4V alloy.

The solution-treated samples were mechanically polished using a standard metallographic procedure and subsequently etched with 20 mL HF + 20 mL $HNO_3$ + 60 mL $H_2O$ to reveal grain boundaries for the microstructure observation. The microstructures of the solution-treated samples were illustrated with an OLYMPUS (Tokyo, Japan) Gx51 optical microscope (OM). The final aged microstructures were etched using Kroll's reagent (10 mL HF +20 mL $HNO_3$ + 70 mL $H_2O$). The microstructures of the aging-treated samples were characterized using a ZEISS (Oberkochen, Germany) SUPRA40 field emission scanning electron microscope (FESEM) at 10 kV under InLens mode with a protective pure nitrogen atmosphere at room temperature.

The phase composition and fraction of the Ti–3.5Al–5Mo–4V alloy were analyzed according to the XRD data. XRD was performed in Bragg–Brentano mode with a step size of 2°/min covering 2θ values in the range of 25–80° on a Bruker D8 Advance (Karlsruhe, Germany) by CuKα radiation (λ = 1.5406 Å) at 40 kV and 40 mA at room temperature. The sample stage of this X-ray diffractometer was rotatable,

so when the grain size was small, the influence of texture could be slightly weakened. Certainly, when the grain size was large, the influence on texture weakening was not obvious. In this work, to further reduce the effect of texture on the volume fraction of the α and β phases, the intensity of the diffraction peak in area was used to calculate the volume fractions of the α and β phases. Moreover, the (110), (200), and (211) diffraction peaks of the β phase, as well as the (100), (002), (101), (102), (110), (103), (112), and (201) diffraction peaks of the α phase, were used to calculate the volume fraction of the α and β phases to reduce the effect of texture on the volume fractions of the α and β phases.

Considering that there were only two phases (β and α) in the Ti–3.5Al–5Mo–4V alloy after aging treatment, the value of the α precipitate fraction ($f_\alpha$) of the alloy was calculated by XRD using a direct comparison methodology [21]:

$$f_\alpha = \frac{1}{1 + \frac{1}{m}\sum_{i=1}^{m}\left(\frac{I_i^\beta}{R_i^\beta}\right)/\frac{1}{n}\sum_{i=1}^{n}\left(\frac{I_i^\alpha}{R_i^\alpha}\right)}, \tag{1}$$

where $m$ and $n$ signify the number of diffraction peaks of the β and α phases, respectively, $I_i$ represents the integrated intensity in area of the $(hkl)$ plane corresponding to the β or α phase, and $R_i$ expresses the scattering factor of the material, which is given by the following equation [21]:

$$R_i = \left(\frac{1}{V_i^2}\right)\left[|F_i|^2 P_i\left(\frac{1 + cos^2 2\theta}{sin^2\theta cos\theta}\right)\right](e^{-2M}), \tag{2}$$

where $V_i$ is the volume of the unit cell, $P_i$ is the multiplicity factor, $F_i$ is the structure factor for the reflecting plane, $e^{-2M}$ is the temperature factor, and $\frac{1 + cos^2 2\theta}{sin^2\theta cos\theta}$ is the angle factor. In this research, the average of all of the diffraction peaks was weighted to ensure accurate calculation results. Taking the sample aged at 500 °C for 4 h after being solution-treated at 850 °C as an example, the values of the parameters $V_i$, $P_i$, $F_i$, $e^{-2M}$, $\frac{1 + cos^2 2\theta}{sin^2\theta cos\theta}$, and $R_i$ are listed in Table 2.

**Table 2.** The values of the parameters $V_i$, $|F_i|^2$, $P_i$, $\frac{1 + cos^2 2\theta}{sin^2\theta cos\theta}$, $e^{-2M}$, and $R_i$ for the sample aged at 500 °C for 4 h after being solution-treated at 850 °C.

| Diffraction Peak / Parameter | α(100) | α(002) | α(101) | α(102) | α(110) | α(103) | α(112) | α(201) | β(110) | β(200) | β(211) |
|---|---|---|---|---|---|---|---|---|---|---|---|
| $V_i$ | 35.15 | 35.15 | 35.15 | 35.15 | 35.15 | 35.15 | 35.15 | 35.15 | 35.39 | 35.39 | 35.39 |
| $|F_i|^2$ | 317.12 | 1194.95 | 866.33 | 226.39 | 780.50 | 521.27 | 650.01 | 160.96 | 1170.97 | 847.86 | 703.35 |
| $P_i$ | 6 | 2 | 12 | 12 | 6 | 12 | 12 | 12 | 12 | 6 | 24 |
| $\frac{1 + cos^2 2\theta}{sin^2\theta cos\theta}$ | 19.13 | 15.80 | 14.19 | 7.62 | 5.12 | 3.94 | 3.50 | 3.40 | 14.88 | 6.43 | 4.05 |
| $e^{-2M}$ | 0.96 | 0.95 | 0.94 | 0.91 | 0.87 | 0.84 | 0.83 | 0.82 | 0.95 | 0.89 | 0.85 |
| $R_i$ | 28.15 | 28.97 | 112.56 | 15.18 | 16.94 | 16.81 | 18.29 | 4.37 | 157.77 | 23.33 | 46.19 |

A microhardness test was carried out on an HVS–1000 digital microhardness measurement machine using a load of 100 g force applied with a diamond indenter for 30 s. Calibration was done on a standardized 458 HV measurement block prior to taking each set of tests. All samples were wet-ground to 2000 grit with waterproof SiC paper and polished for testing. There were at least seven points measured for each specimen, and the average value was taken to diminish measurement error (the measured error was within 5%).

## 3. Results and Discussion

### 3.1. Microstructure and Phase Composition of the Samples after Solution Treatment

The optical micrographs of the Ti–3.5Al–5Mo–4V alloy after solution treatment at 850 and 1050 °C are shown in Figure 2. It can be seen from Figure 2a,b that the microstructure presented a matrix of metastable β grains of approximately equiaxed morphology, with mean grain sizes of 112 and 289 μm, respectively.

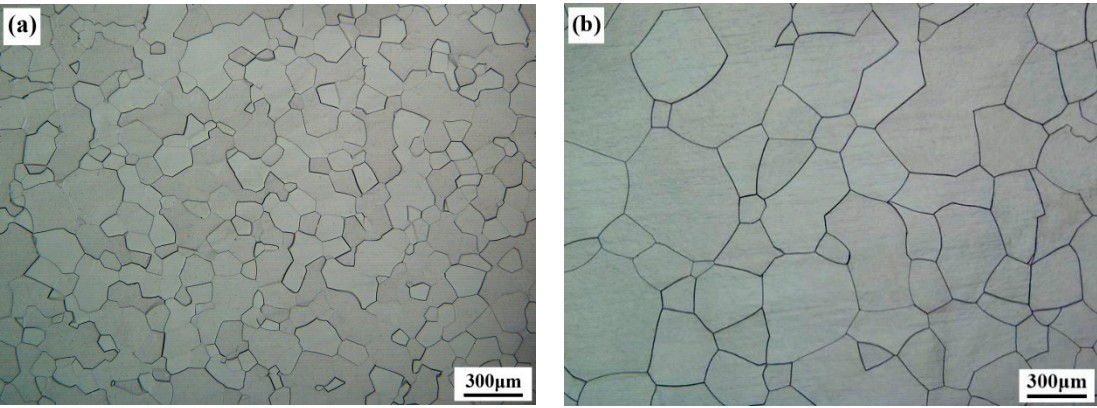

**Figure 2.** OM image of the initial water-quenched microstructure of the Ti–3.5Al–5Mo–4V alloy formed by a matrix of metastable β grains of approximately equiaxed morphology. (**a**) 850 °C and (**b**) 1050 °C.

A more detailed analysis of the XRD patterns was performed after solution treatment. Figure 3 shows the XRD diffractograms of the samples after being solution-treated at temperatures of 850 and 1050 °C. There were only three β phase diffraction peaks ($β_{(100)}$, $β_{(200)}$, and $β_{(211)}$) in the XRD image. Some researchers [14] suggested that an athermal ω phase could form during the cooling process of Ti–5Al–5Mo–5V–3Cr alloy from a high solution temperature. However, as we know, low volume fractions (less than 3% or 5%) of some phases could not be detected by XRD. According to Figure 3, the peak of the ω phase was not observed. Therefore, the microstructure was mainly a single β phase for the Ti–3.5Al–5Mo–4V alloy after solution treatment. Even though the ω phase may have been present, its volume fraction must have been very low.

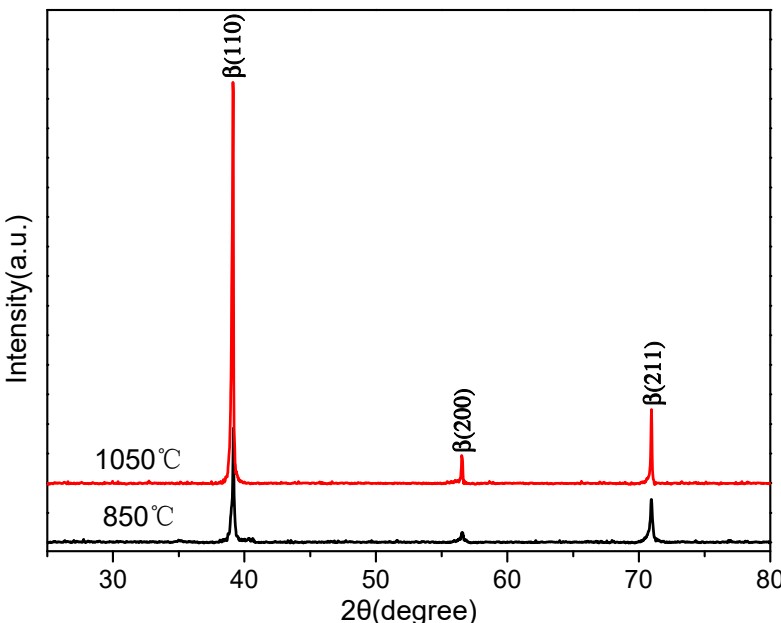

**Figure 3.** XRD patterns of the alloy solution-treated at 850 and 1050 °C.

### 3.2. Phase Compositions and Microstructures of the Samples after Aging Treatment

XRD is not normally used to quantitatively analyze α and α + β titanium alloys, mainly because a martensite transition may occur during quenching and it is difficult to discriminate between the α and α′ (martensite) phases in these alloys [33]. However, for metastable β-titanium alloys, the initiation temperature of the martensite transition is generally below room temperature. Hence, the β and α phases can be easily detected and distinguished according to XRD. Furthermore, the β → α transition

belongs to the univariate type in titanium alloys. The proportions and equilibrium contents between the α and β phases depend on the heat treatment temperature, holding time, and cooling rate.

In order to distinctly research the β → α phase transition in Ti–3.5Al–5Mo–4V alloy under isothermal aging treatments, the different aging times of the samples were analyzed by XRD. The typical XRD diffractograms are displayed in Figure 4.

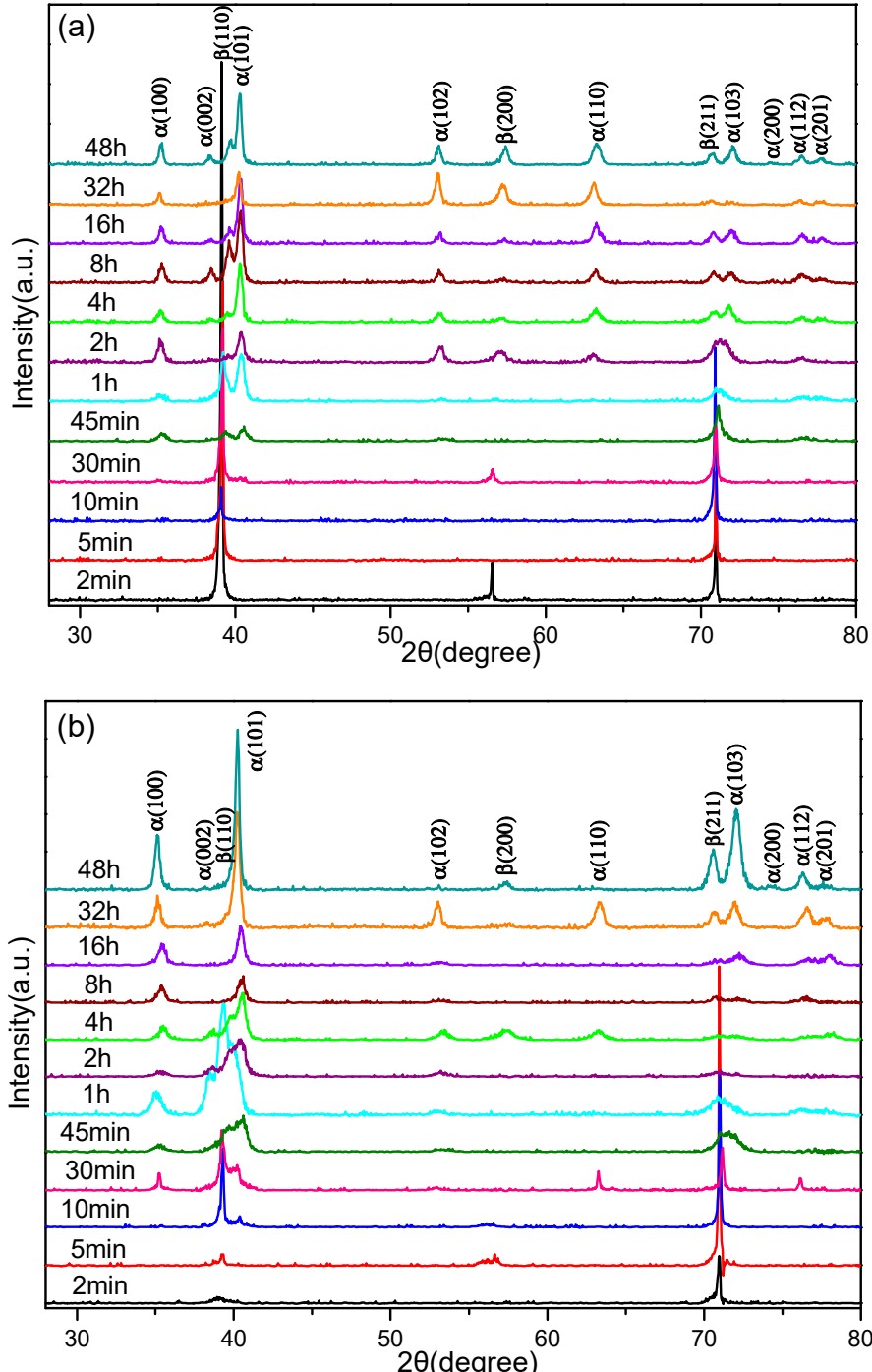

**Figure 4.** XRD patterns of the Ti–3.5Al–5Mo–4V alloy isothermally aged at 500 °C for different times after being solution-treated at (**a**) 850 °C and (**b**) 1050 °C.

As shown in Figure 4a, there was no α diffraction peak in the first 10 min, which provided evidence of the microstructure of the Ti–3.5Al–5Mo–4V alloy fully remaining in the β phase after aging

treatment for the samples that were solution-treated at 850 °C. When the aging time was extended to 30 min, the diffraction peaks of α precipitation began to appear alongside the β diffraction peaks, on the XRD diffractograms. Prolonged aging time at this temperature also gave rise to a great quantity of α phase. When the aging time reached 10 min, the α diffraction peaks were detected on the XRD pattern (Figure 4b) of the samples that were solution-treated at 1050 °C, while only the β diffraction peaks emerged for the samples that were solution-treated at 850 °C. This phenomenon demonstrated that the coarse β grains produced by a higher solution temperature were more advantageous for the β → α phase transition. When the aging time increased from 10 min to 8 h, the quantity of the α phase increased rapidly, which reflected the Ti–3.5Al–5Mo–4V alloy displaying a rapid aging response. Long-term aging for 16 h showed that the microstructure was basically composed of the α phase. Furthermore, the precipitation content of the α phase in the Ti–3.5Al–5Mo–4V alloy reached an equilibrium value, regardless of the solution temperature.

Figures 5 and 6 show SEM micrographs of Ti–3.5Al–5Mo–4V alloy that was isothermally aged for 45 min after solution treatment. Figure 5a,b shows that there was merely a small quantity of acicular α precipitate generated on the β matrix. A large quantity of α phase precipitated on the β grain boundaries after being aged at 500 °C for 45 min, revealing that the β grain boundaries may have been the dominant and preferential nucleation sites of the α precipitation. As seen from Figure 5, in addition to grain boundary α, "side plates" also nucleated at the grain boundary and grew toward the interior of the grain. "Side plates" have the ability to consume the β stabilizing element in the grain boundary and weaken the growth of grain boundary α, as well as reduce the size and break the continuity of grain boundary α [34]. Therefore, "side plates" promote grain boundary strengthening to improve the strength level of the alloy [35]. Similar behavior was also observed in previous studies on Ti–55531 and Ti–3Al–15Mo–3Nb–0.2Si alloys [11,36]. Moreover, the density of the α phase that precipitated from the grain boundaries in the alloy increased with the rise in solution temperature. This phenomenon was mainly due to the fact that the solubility of the alloying elements increased with increasing solution temperature. Thus, a higher supersaturation in the solute was in the β phase during quenching, leading to a higher driving force for α precipitation. In general, when the driving force of the precipitation was large enough, high-density precipitation was generated [7]. Therefore, the precipitation of the α phase was denser in the studied alloy at a higher solution temperature.

When the aging time was short, an elongated acicular α phase precipitated first. After prolonging the aging time, a relatively short acicular α precipitation interleaved with the elongated acicular α phase that precipitated first, as shown in Figure 6. The acicular α phases in different directions were interwoven with the growth of the acicular α phase (Figure 7c). Finally, the closely distributed α phase was formed (Figure 7e).

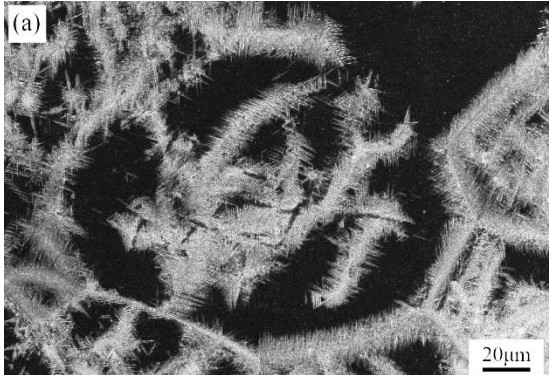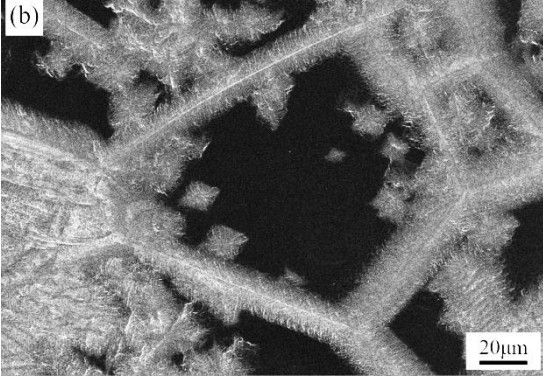

**Figure 5.** SEM micrograph of the alloy that was isothermally aging-treated at 500 °C for 45 min after being solution-treated at (**a**) 850 °C and (**b**) 1050 °C.

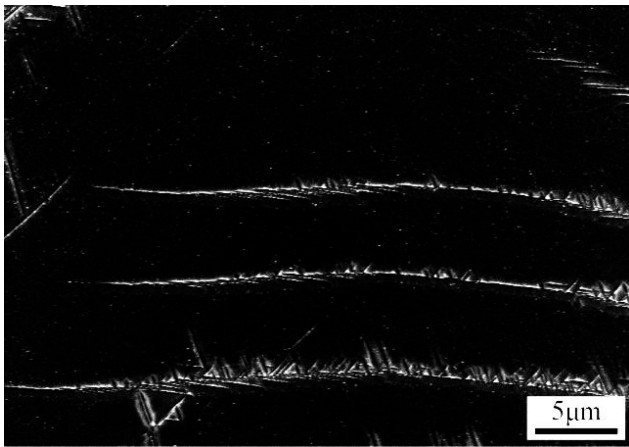

**Figure 6.** SEM micrograph of Ti–3.5Al–5Mo–4V alloy isothermally aged at 500 °C for 45 min after being solution-treated at 850 °C.

**Figure 7.** SEM micrographs of Ti–3.5Al–5Mo–4V alloy that was solution-treated at (**a**,**c**,**e**) 850 °C and (**b**,**d**,**f**) 1050 °C, then isothermally aging-treated at 500 °C for (**a**,**b**) 45 min, (**c**,**d**) 2 h, and (**e**,**f**) 16 h.

Furthermore, it was useful to research the morphological evolution of the precipitation of the $\alpha$ phase with the aging time under a constant aging temperature. Figures 7 and 8 show the micrographs of Ti–3.5Al–5Mo–4V alloy aged at 500 °C for different times after solution treatment. Just a small quantity of the acicular $\alpha$ phase was generated after aging at 500 °C for 45 min (Figure 7a,b). The value of the $\alpha$ precipitate fraction increased when the aging time was longer (Figure 7c–f). In addition, the value of the $\alpha$ precipitate fraction was higher in the alloy at a higher solution temperature under the same aging treatment.

However, the size of the $\alpha$ precipitation seemingly did not increase linearly after a short aging time, which was mainly ascribed to the inhomogeneous precipitation of the $\alpha$ phase at the beginning. As seen from Figure 8, the size of the $\alpha$ phase gradually increased with the aging time when it was longer. Furthermore, the $\alpha$ precipitate size was smaller in the Ti–3.5Al–5Mo–4V alloy at a higher solution temperature after a given aging time, implying that the coarse $\beta$ grain was not conducive to the growth of the $\alpha$ phase that precipitated in the alloy. Also, this may be due to there being fewer residual stresses and defects in the coarse grain than in the fine grain, which restrained element diffusion in the aging-treatment process [1].

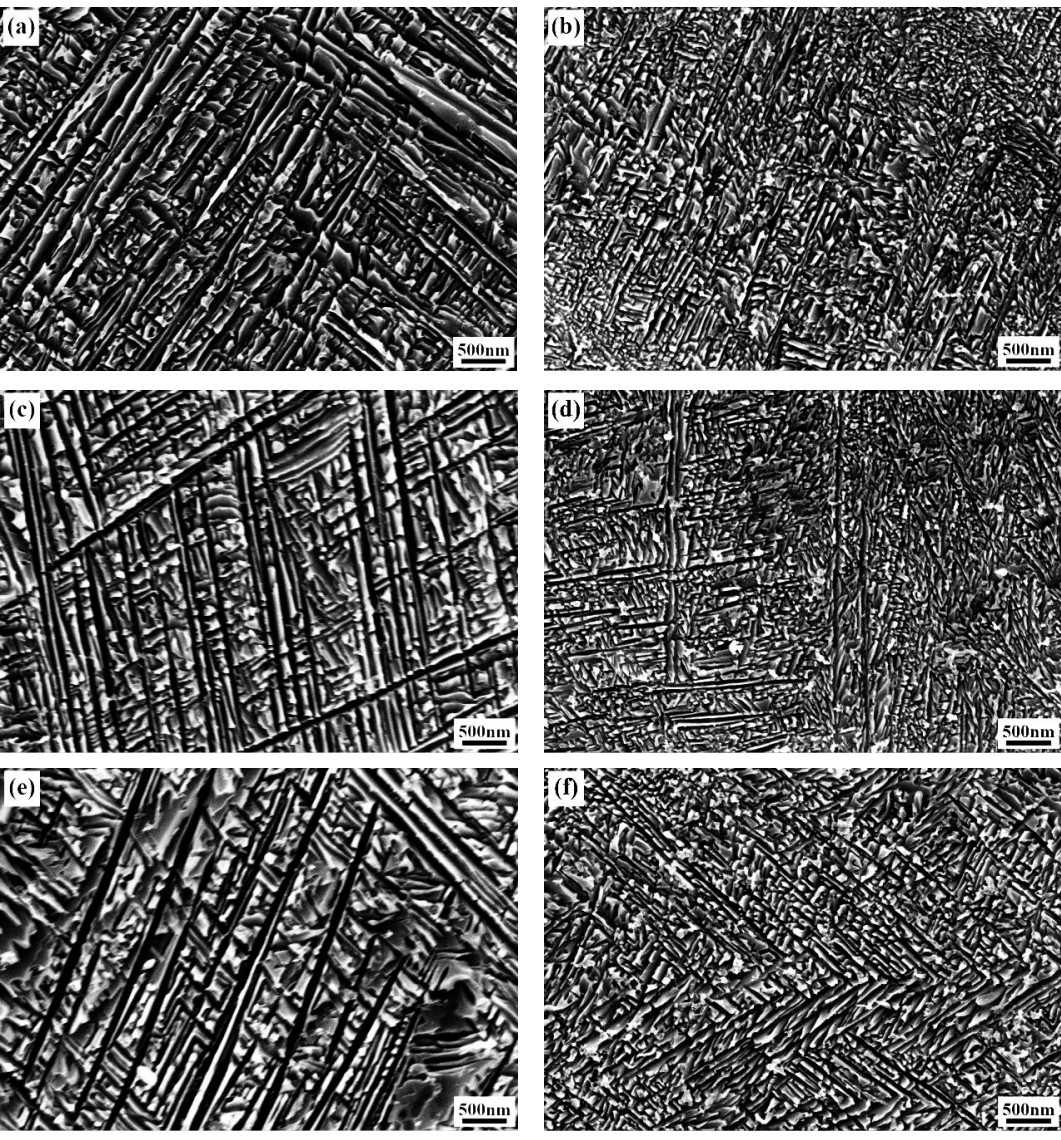

**Figure 8.** SEM micrographs of Ti–3.5Al–5Mo–4V alloy solution-treated at (**a,c,e**) 850 °C and (**b,d,f**) 1050 °C, then isothermally aging-treated at 500 °C for (**a,b**) 8 h, (**c,d**) 16 h, and (**e,f**) 48 h.

### 3.3. Kinetics of β → α Phase Transition

It was apparent that the size and amount of the α phase that developed in the β matrix was a function of the aging time for the Ti–3.5Al–5Mo–4V alloy solution-treated at 850 and 1050 °C, as seen from Figures 7 and 8. In order to quantitatively describe the relationship between the volume fraction of the α phase and aging time, the value of the α precipitate fraction in the Ti–3.5Al–5Mo–4V alloy under different aging treatments was calculated using a direct comparison methodology by Equations (1) and (2). In order to obtain the volume fraction of the α phase more accurately, typical XRD patterns of Ti–3.5Al–5Mo–4V alloy aged at 500 °C for different times after solution treatment were fitted, as shown in Figure 9. Good consistency was obtained in the results of the measured and fitted data.

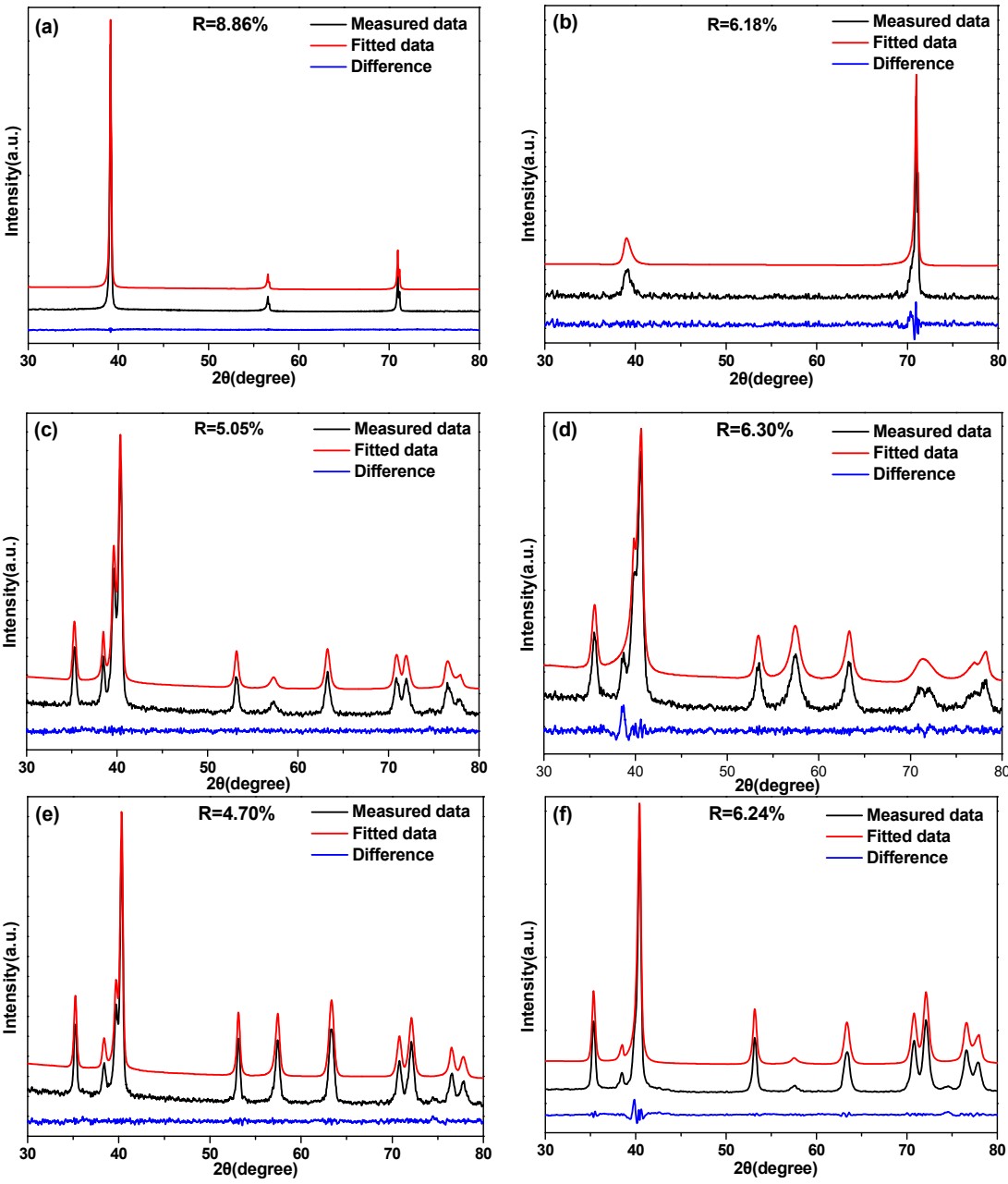

**Figure 9.** The profile fitting results of the XRD patterns of the Ti–3.5Al–5Mo–4V alloy solution-treated at (**a,c,e**) 850 °C and (**b,d,f**) 1050 °C, then isothermally aging-treated at 500 °C for (**a,b**) 2 min, (**c,d**) 4 h, and (**e,f**) 32 h.

Figure 10 shows the variation in the values of the α precipitate fraction calculated using the JMAK theory and measured by XRD in Ti–3.5Al–5Mo–4V alloy aged at 500 °C for different times after solution treatment. The results demonstrated that the value of the α precipitate fraction increased sharply at first. This increase was due to an increase in the quantity of the α phase nuclei, as well as the growth of the existing α precipitates [14]. Then, it increased slowly as the aging time increased. The distance between the particles possibly decreased gradually as the value of the α precipitate fraction increased. Moreover, the strong strain elastic interactions between the precipitates resulted in growth rate reduction [37]. As the aging time further increased, the fraction of the α precipitate reached an equilibrium value. Furthermore, the equilibrium value of the precipitation of α phase was higher in the Ti–3.5Al–5Mo–4V alloy at a higher solution temperature, mainly resulting from the higher solution temperature improving the solubility of the alloying elements and reducing the stability of the β matrix, which brought about a higher driving force of the precipitation. As a result, it was conducive to the precipitation of the α phase after the aging treatment from the perspective of the phase transition kinetics.

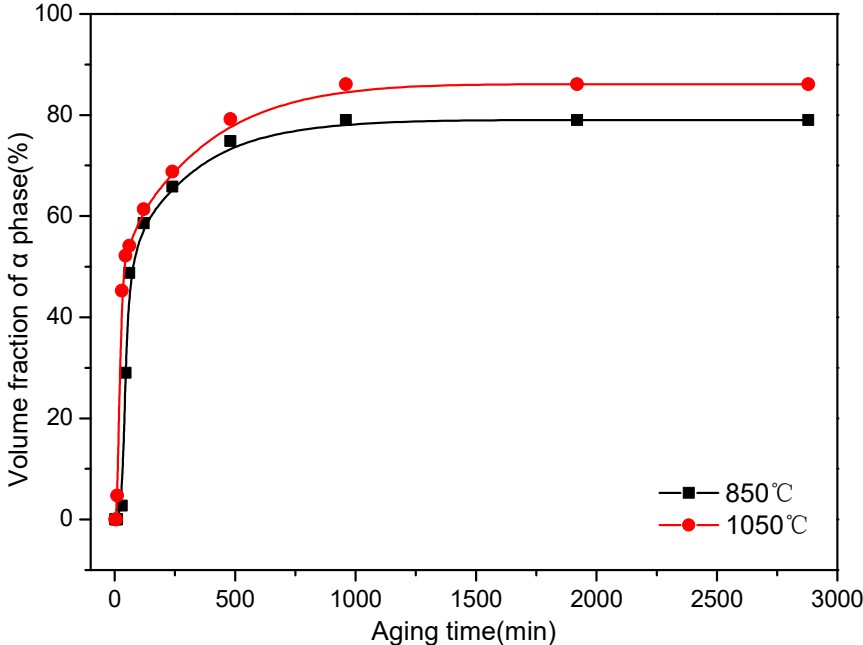

**Figure 10.** Variation in the volume fraction of α phase at 500 °C isothermal treatment with different aging times for the Ti–3.5Al–5Mo–4V alloy after solution treatment at 850 and 1050 °C. Points with different symbols represent measured values and lines represent simulated results.

According to the Avrami theory, the experimental data obtained by XRD were analyzed using the Johnson–Mehl–Avrami–Kolmogorov (JMAK) equation [38–41]. The relationship between the value of the α precipitate fraction and aging time were also determined from the isothermal phase transformation kinetics using the Avrami theory and the JMAK equation. For the isothermal transformation, the equation had the following common form:

$$f = 1 - exp(-kt^n),\tag{3}$$

where $f$ is the product value of the precipitate fraction, $k$ is a constant of the reaction rate, $t$ is the transformation time, and $n$ corresponds to the Avrami index, which reflects the nucleation and growth mechanism.

One bold and novel attempt in this study was to model the β → α phase transition kinetics on the basis of the classic JMAK theory for Ti–3.5Al–5Mo–4V alloy aged at 500 °C. In order to achieve the research objective, a rewritten form of the above-mentioned equation was produced [42]:

$$\frac{f_\alpha(t)}{f_\alpha^{max}} = 1 - exp(-kt^n), \tag{4}$$

where $f_\alpha(t)$ is the value of the $\alpha$ precipitate fraction in the sample aged at 500 °C for aging time $t$, $f_\alpha^{max}$ is the maximum (equilibrium) value of the $\alpha$ precipitate fraction in the transition at 500 °C, and $\frac{f_\alpha(t)}{f_\alpha^{max}}$ is the degree of the transition. However, it must be noted that the JMAK equation expresses the transition from the beginning, without considering the preprocessing and incubation period. Consequently, the time $t$ should not be regarded as the absolute time but should be considered as the time corresponding to the beginning of the transition. Therefore, a simple subtraction should be made, i.e., subtracting the incubation time of the transition at 500 °C.

The derived JMAK parameters were used to calculate the phase transition kinetics of the alloy aged at 500 °C. According to Equation (4), the experimental data obtained from the XRD were analyzed with logarithmic plots, that is, the relationship between $ln\left[-ln\left(1 - \frac{f_\alpha(t)}{f_\alpha^{max}}\right)\right]$ and $ln(t)$. The slope of the resulting straight line corresponded to $n$ (Avrami exponent), and the value of $k$ was obtained on the base of the intercept. Such a plot is shown in Figure 11 regarding the results obtained from the XRD study.

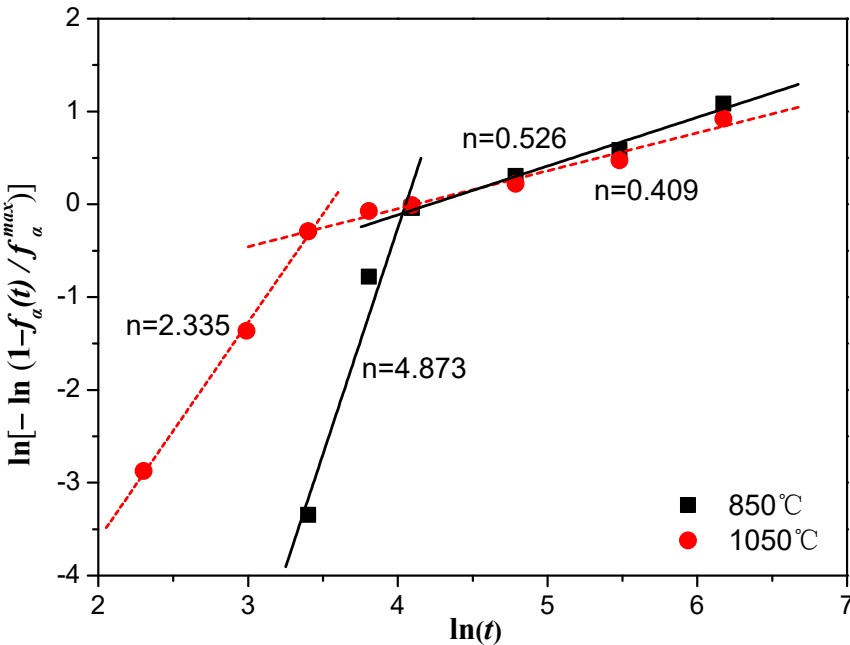

**Figure 11.** Plot of $ln[-ln(1 - f_\alpha(t)/f_\alpha^{max})]$ versus $ln(t)$ for Ti–3.5Al–5Mo–4V alloy isothermally aged at 500 °C after being solution-treated at different temperatures.

The experimental data were obviously composed of two parts; thus, they could be fitted with two divided lines. The slopes of the fitting straight line of the data of the alloy that was solution-treated at 850 and 1050 °C were 4.873 and 2.335 in the first stage of the phase transition. Nevertheless, as the aging time increased, the line slopes of the fitting straight line of the alloy solution-treated at 850 and 1050 °C showed an obvious downward tendency, and the values were reduced to about 0.526 and 0.409, respectively. Furthermore, it is interesting to note that the Avrami index ($n$) was relatively higher in the Ti–3.5Al–5Mo–4V alloy at a lower solution temperature, while the reaction rate constant ($k$) was the opposite. The slope $n$ had a bearing on the nucleation and growth mechanism of the $\alpha$ precipitation

that was transformed in the alloy. The significant change in the slope of the straight line implied that the phase transition mechanism changed in the process of the transition. A similar observation was reported in previous research regarding β21s and β-CEZ alloys [18,33]. The mechanism of the phase transition in the first stage was a mixed transformation type (homogeneously nucleated and acicular-grown α structure, and grain boundary-nucleated and grown α precipitate), while the second stage involved the growth of fine α precipitates controlled by slow diffusion [42]. This was different than Ti15-3 alloy [24], the experimental results of which were fitted perfectly with one straight line and an Avrami index of 0.37. Therefore, the phase transition mechanism of the Ti15-3 alloy was considered to be the growth of fine α precipitates controlled by slow diffusion.

The JMAK parameters (*n* and *k*) and the correlation coefficients (*r*) of the resulting linear fit curves of the Ti–3.5Al–5Mo–4V alloy aged at 500 °C after being solution-treated at 850 and 1050 °C are listed in Table 3. The calculated values were in good agreement with the experimental values, demonstrating that β → α phase transition kinetics of Ti–3.5Al–5Mo–4V alloy under isothermal conditions after solution treatment at different temperatures can be predicted by the JMAK theory.

**Table 3.** The Johnson–Mehl–Avrami–Kolmogorov (JMAK) parameters (*n* and *k* ) and the correlation coefficients (*r* ) of linear fits at 500 °C obtained by the plots of $ln[-ln(1 - f_\alpha(t) / f_\alpha^{max}]$ versus $ln(t)$ for Ti–3.5Al–5Mo–4V alloy under isothermal transformation conditions after solution treatment at 850 and 1050 °C.

| Solution Treatment Temperature, °C | *n* | *k* | *r* |
|---|---|---|---|
| 850 | 4.87286 ± 0.03922 | $2.65243 \times 10^{-9}$ | 0.97801 |
| | 0.52571 ± 0.04553 | 0.108961573 | 0.99258 |
| 1050 | 2.33497 ± 0.07564 | 0.000254088 | 0.99894 |
| | 0.40872 ± 0.02978 | 0.185691237 | 0.98955 |

### 3.4. Age-Hardening Behavior

Microhardness testing was applied to characterize the mechanical properties of the Ti–3.5Al–5Mo–4V alloy after aging treatment, providing a considerable amount of appropriate information to effectively evaluate the age-hardening of a great deal of samples of an alloy over a wide range of temperatures and a long period of time. Microhardness is essentially related to the yield strength of a material to estimate or evaluate its strength [43,44].

To better understand the influence of the α precipitate on the mechanical properties of Ti–3.5Al–5Mo–4V alloy during aging treatment, the curves of the Vickers hardness and aging times of the studied alloy are graphically shown in Figure 12. The hardness of the original sample that was solution-treated at 850 °C was relatively lower than at 1050 °C. This may be ascribed to the fact that the distribution of the alloying elements was more homogeneous as the solution temperature increased, therefore resulting in a stronger solution strengthening effect. On the other hand, greater quenching stress in the interior of the grains was produced when the sample was quenched from a higher solution temperature to room temperature, which gave rise to more internal defects. The same phenomenon occurred when the aging time increased. This can be attributed to the fact that the α precipitate formed by the aging treatment after being solution-treated at 850 °C was relatively easy to coarsen and the α precipitate fraction was relatively lower, resulting in the hardness value always being lower than it was at 1050 °C.

Broadly, the quantity of the α phase that was transformed was a function of the aging time; hence, there was clearly a strong functional relationship between the hardness and the aging time. When the aging time was short, the hardness value of the sample changed very little at the beginning stage of the aging treatment. This behavior was attributed to the precipitation of α phase requiring a certain incubation time and the small amount of the α phase at the beginning of precipitation, which had little effect on the hardness value of the sample. When the aging time increased, the hardness of the sample increased sharply due to the α phase being a solid solution with a hexagonal closely packed

crystal structure, which limited the number of operative slip systems. On the other hand, the great quantity of fine α phase precipitates in the β matrix provided more α/β interfaces, which acted as dislocation barriers and effectively hindered the dislocation movement [45]. Therefore, a kind of precipitation hardening was generated. Peak hardness was attained after aging for about 8 h, regardless of the solution temperature. When the sample was solution-treated at 850 °C, the hardness value was near 500 HV, and the hardness value was about 640 HV when solution-treated at 1050 °C. After the hardness value of the sample reached a plateau, it began to show a downward trend, which was mainly attributed to the precipitated α phase becoming thicker and longer with further increases in aging time (Figure 8). Similar results were also observed in previous research regarding Ti–2Al–9.2Mo–2Fe alloy when it was aged at 500–600 °C [46]. This also reflected a relative under-aging, peak aging, and over-aging period.

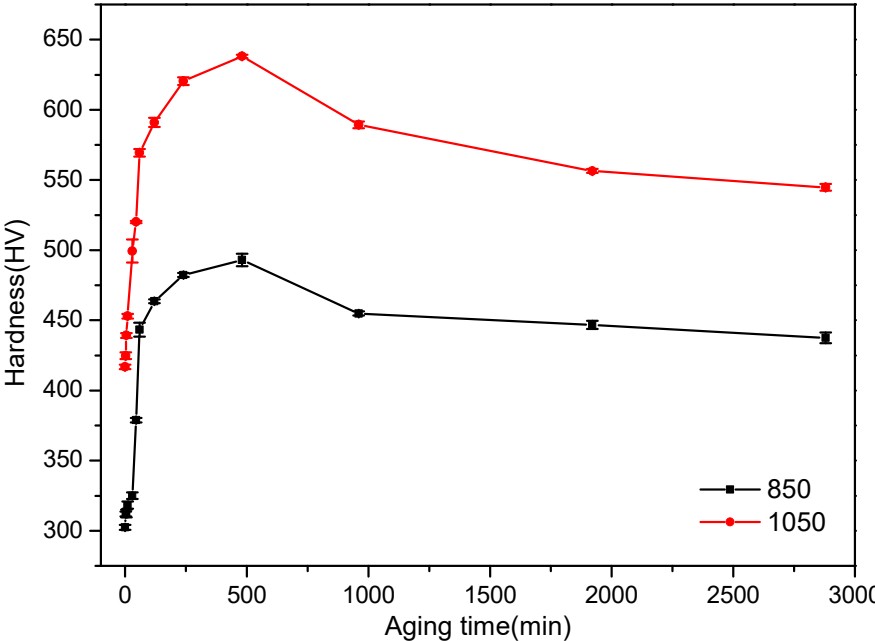

**Figure 12.** Microhardness of the samples aged at 500 °C as a function of aging time for Ti–3.5Al–5Mo–4V alloy after solution treatment at 850 and 1050 °C.

## 4. Conclusions

The β → α phase transition kinetics under isothermal conditions were investigated in the Ti–3.5Al–5Mo–4V alloy with two different grain sizes based on XRD. The main results of this investigation can be summarized as follows:

1. In the early stage, the isothermally-aged α phase precipitated at the initial β grain boundaries. The value of the α precipitate fraction increased sharply at first and then increased slowly with increasing aging time, finally reaching equilibrium. The value of the α precipitate fraction was higher in the alloy aged for the same time at a higher solution temperature, while the size of the α precipitate was smaller in the alloy at a higher solution temperature.
2. The isothermal β → α phase transition kinetics under isothermal-aging treatments were modeled in the theoretical framework of the JMAK theory. The kinetic parameters of JMAK deduced different transformation mechanisms in the process of the transition. The mechanism of the first-stage phase transition was dominated by mixed transformation mechanisms (homogeneously nucleated and acicular-grown α structure, and grain boundary-nucleated and grown α precipitate), while the second stage involved the growth of fine α precipitates controlled by slow diffusion. The Avrami index (*n*) was higher for Ti–3.5Al–5Mo–4V alloy at a lower solution temperature,

while the reaction rate constant (*k*) showed the opposite. A very good correspondence between calculated and experimentally measured values was found.

3.  As the aging time was prolonged, the hardness of the Ti–3.5Al–5Mo–4V alloy increased sharply. After the hardness of the alloy reached a plateau, it began to decline. The hardness of the alloy was always higher at a higher solution temperature.

**Author Contributions:** Conceptualization, S.X.; methodology, formal analysis, writing—original draft, writing—review and editing, P.G.; software and data curation, Y.T.; resources, X.J.; supervision, validation, project administration and funding acquisition, S.X. All authors have read and agreed to the published version of the manuscript.

**Funding:** This work was financially supported by the National Natural Science Foundation of China (Grant No. 51974097, 51774103and 51661006), the Program of "One Hundred Talented People" of Guizhou Province (Grant No. 20164014), the Science and Technology Project of Guizhou Province (Grant No. 20175656, 20175788, 20191414 and 20192162), and the Program for Innovation Research Team of Guizhou Province Education Ministry (Grant No. 2016021).

**Conflicts of Interest:** The authors declare no conflict of interest.

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
