# Peer review of "Studies on the β → α Phase Transition Kinetics of Ti–3.5Al–5Mo–4V Alloy under Isothermal Conditions by X-ray Diffraction"

_metals, doi:10.3390/met10010090_

Round 1

Reviewer 1 Report

The beta->alpha phase transitions kinetics of a Ti alloy are investigated during isothermal aging primarily by X-ray diffraction and supported by SEM, hardness measurements, and JMAK modelling. Quantitative phase fractions were derived for two grain sizes as function of annealing time. The JMAK analysis indicates two distinct transformation mechanisms which are consistent with SEM micrographs. Hardness is also measured and correlated to the alpha precipitation. I consider the presented observations of sufficient interest for publication.

Traditional experimental and modelling approaches are employed. The choice of the techniques is appropriate and the study appears to be in general technically sound. I just wonder about the apparently erratic intensity of alpha peaks in Figure 4b (1050C ). The intensity seems to be strong at 4h aging but very weak at 2h and 8h, and very strong at 32h. This appears in contrast to the smooth increase of volume fraction shown in Figure 8. Is this a scaling effect of the individual curves due to insufficient grain statistics (large grain size)?

Figure 10 was missing in the review manuscript, so I cannot judge the conclusions from the dependency of hardness upon annealing time.

Author Response

Response to the Reviewer1’s Comments

We would like to thank the Reviewer for his/her helpful comments. We have followed all the suggestions and made revisions accordingly. The list below gives the details of the changes that we have made in the revised manuscript.

Point 1: Traditional experimental and modelling approaches are employed. The choice of the techniques is appropriate and the study appears to be in general technically sound. I just wonder about the apparently erratic intensity of alpha peaks in Figure 4b (1050oC). The intensity seems to be strong at 4h aging but very weak at 2h and 8h, and very strong at 32h. This appears in contrast to the smooth increase of volume fraction shown in Figure 8. Is this a scaling effect of the individual curves due to insufficient grain statistics (large grain size)?
Response 1: Due to the existence of crystal texture during heat treatment under different aging time, the intensity of diffraction peak in height is different at different diffraction angle of 2θ. In this present work, in order to weak the effect of texture on the volume fraction of α and β phase, the intensity of diffraction peak in area was used to calculate the volume fraction of α and β phase. Moreover, the (110), (200) and (211) diffraction peaks of β phase as well as the (100), (002), (101), (102), (110), (103), (112) and (201) diffraction peaks of α phase were used to calculate the volume fraction of α and β phase to reduce the effect of texture on the volume fraction of α and β phase.
At the same time, they have been further supplemented, please seen the section of 2 in the manuscript.

Point 2: Figure 10 was missing in the review manuscript, so I cannot judge the conclusions from the dependency of hardness upon annealing time.
Response 2: According to the reviewer’s suggestion, Figure 10 has been supplemented, please seen the section of 3.4 in the manuscript.

Figure 10. Micro hardness of the samples aged at 500°C as a function of aging time for the Ti–3.5Al–5Mo–4V alloy after solution treated at 850°C and 1050°C.
Point 3: Moderate English changes required.
Response 3: According to the reviewer’s suggestion, we have carefully read the manuscript and made some revisions.

Reviewer 2 Report

Dear authors,

Defining the kinetics of the beta-alpha transition in titanium alloys would be a good work. However, the manuscript lacks support for the arguments presented and an extensive English review is necessary.

The authors use the fraction of the alpha precipitates to describe the kinetics of the transition. However, the XRD measurements were performed within the 2 theta range of 25 - 80 degrees. As far as I am concerned, alpha and beta phases intense peaks up to 130 degrees. Why did the authors not measure at a higher 2 theta? There is no information on the fitting of the XRD patterns. Since the authors have the XRD diffractograms and, if there is no texture, why not doing the Rietveld refinement to have the phase percentage? Moreover, even though the authors mention that the sample was rotated to decrease the influence of the texture, they did not provide any analysis supporting this hypothesis. Rotation itself does decrease the influence of the texture. However, the crystallite sizes might be big and, in this case, there would be not enough random orientation of the grains to give enough statistics for all the crystal planes. Figure 4a, for example, shows the beta (200) peak only on a few patterns (2 min, 30 min, 2 h, etc), while it is not visible in others (5 min, 10 min, 45 min, etc). Could it be because of texture? Alpha peaks (100), for example, does not increase the intensity consistently. The diffraction peak is intense in 45 min, then it decreases the intensity in 1 h, increases again for 2 h, decreases for 4 h.

Author Response

Response to the Reviewer2’s Comments

We would like to thank the Reviewer for his/her helpful comments. We have followed all the suggestions and made revisions accordingly. The list below gives the details of the changes that we have made in the revised manuscript.

Point 1: The authors use the fraction of the alpha precipitates to describe the kinetics of the transition. However, the XRD measurements were performed within the 2 theta range of 25 - 80 degrees. As far as I am concerned, alpha and beta phases intense peaks up to 130 degrees. Why did the authors not measure at a higher 2 theta?

Response 1: The strong peak of titanium alloy is mainly distributed within 80°. When 2θ is greater than 80°, the diffraction peak is weak and can be neglected during the calculation process of the volume fraction of α and β phase [1-5]. Therefore, the XRD measurements were performed within the 2θ range of 25 – 80° in this study.

[1] Chen F.W.; Xu G.L.; Zhang X.Y.; Zhou K.C.; Cui Y.W. Effect of α morphology on the diffusional β ↔ α transformation in Ti–55531 during continuous heating: Dissection by dilatometer test, microstructure observation and calculation. J. Alloys Compd., 2017, 702, 352-365.

[2] Xu T.W.; Zhang S.S.; Zhang F.S.; Kou H.C.; Li J.S. Effect of ω-assisted precipitation on β→α transformation and tensile properties of Ti–15Mo–2.7Nb–3Al–0.2Si alloy. Mater. Sci. Eng. A 2016, 654, 249-255.

[3] Stella P.; Giovanetti I.; Masi G.; Leoni M.; Molinari A. Microstructure and microhardness of heat-treated Ti–6Al–2Sn–4Zr–6Mo alloy. J. Alloys Compd. 2013, 567, 134-140.

[4] R Santhosh.; Geetha M.; Saxena V.K.; Nageswararao M. Studies on single and duplex aging of metastable beta titanium alloy Ti–15V–3Cr–3Al–3Sn. J. Alloys Compd. 2014, 605, 222-229.

[5] Sun F.; Li J.S.; Kou H.C.; Tang B.; Chen Y.; Chang H.; Cai J.M. β phase transformation kinetics in Ti60 alloy during continuous cooling. J. Alloys Compd. 2013, 576, 108-113.

Point 2: There is no information on the fitting of the XRD patterns.

Response 2: In the process of calculating the volume fraction of α and β phase, the fitting of the XRD patterns was performed.

Point 3: Since the authors have the XRD diffractograms and, if there is no texture, why not doing the Rietveld refinement to have the phase percentage?

Response 3: It is really a good method to get the volume fraction of α and β phase by the Rietveld refinement. However, it is very difficult to refine the diffraction peak of α and β phase because of the existence of crystal texture during heat treatment under different aging time.

Point 4: Moreover, even though the authors mention that the sample was rotated to decrease the influence of the texture, they did not provide any analysis supporting this hypothesis. Rotation itself does decrease the influence of the texture. However, the crystallite sizes might be big and, in this case, there would be not enough random orientation of the grains to give enough statistics for all the crystal planes.

Response 4: Choosing the rotatable stage is to make the sample rotate along its surface normal axle during the θ scanning to increase the chances that the normal axle of a crystal plane to cross the diffraction plane on the X-ray diffractometer [1]. In this case, the diffraction intensity of each crystal plane is the average intensity. So when the grain size is small, the influence of texture can be slightly weakened. Certainly, when the grain size is large, the influence on texture weakening is not obvious. In this present work, to further reduce the effect of texture on the volume fraction of α and β phase, the (110), (200) and (211) diffraction peaks of β phase as well as the (100), (002), (101), (102), (110), (103), (112) and (201) diffraction peaks of α phase were used to calculate the volume fraction of α and β phase.

At the same time, they have been further supplemented, please seen the section of 2 in the manuscript.

[1] Guo Z.Q.; Fu T.; Wang N.; Fu H.Z. A simple XRD method for determining crystal orientation and its distribution. J. Inorg. Mater. 2004, 17, 460-464.

Point 5: Figure 4a, for example, shows the beta (200) peak only on a few patterns (2 min, 30 min, 2 h, etc), while it is not visible in others (5 min, 10 min, 45 min, etc). Could it be because of texture? Alpha peaks (100), for example, does not increase the intensity consistently. The diffraction peak is intense in 45 min, then it decreases the intensity in 1 h, increases again for 2 h, and decreases for 4 h.

Response 5: An obvious difference in the diffraction peak intensity of α and β phase was observed at different aging time, which is attributed to the existence of crystal texture during heat treatment under different aging time. In this present work, in order to weak the effect of texture on the volume fraction of α and β phase, the intensity of diffraction peak in area was used to calculate the volume fraction of α and β phase. Moreover, the (110), (200) and (211) diffraction peaks of β phase as well as the (100), (002), (101), (102), (110), (103), (112) and (201) diffraction peaks of α phase were used to calculate the volume fraction of α and β phase to reduce the effect of texture on the volume fraction of α and β phase.

Point 6: Extensive editing of English language and style required

Response 6: According to the reviewer’s suggestion, we have carefully read the manuscript and made some revisions.

Round 2

Reviewer 2 Report

Dear authors,

It is clear that you did some improvements to the article. However, I would suggest an extensive English correction, including the type of language used. Just adding the article is not an extensive correction. I can give a few examples of English corrections I would suggest:

line 14

I believe "dependence", instead of "function", would represent better the idea here. Also, "on the basis of" should be "based on"

lines 15 - 16

Please, reformulate the phrase "The value of .... aging time. Finally, it reached an equilibrium value."

line 25

"it began to decline"

line 34

I believe the word "marvelous" is not a good adjective here.

line 38

Please, reformulate "... is such a new titanium alloy"

line 49

Please, reformulate "alpha phase is hard and un-deformable particle to hinder"

For instance, it is possible to have at least two meanings for this phrase:

alpha phase is hard, and undeformed particles hinder the dislocation movement

or

alpha phase is hard and undeformed, what hinders the dislocation movement?

line 123

X-ray diffraction

A few times the authors use "on this occasion", which is not a good choice of expression for the case.

The word "spectrum" is used instead of "diffractogram". A diffractogram is not a spectrum.

"In view of the fact that" is verbose and could be, in most places, completely removed.

What is "put forward" supposed to mean in the manuscript? I believe the expression should be modified by something else - presented? Introduced?

What is "As is known to all" supposed to mean?

What does "a very instructive function" means?

Most of the units are not following the technical and scientific writing rules.

There are several other examples of English corrections to be performed that I will not add here.

Other corrections not related to language:

The introduction contains a mixture of "beta-titanium", "beta-Ti", "Ti" alloys. Please, double-check if those alloys are really beta or just Ti.

line 40

Would you please add the unit to 10.3?

line 41

"being different from traditional metastable beta-Ti alloys"

How different? Different in which aspects?

table 1

Why different elements have different accuracy if, I suppose, they were measured under the same conditions in the same equipment?

Please, add more technical information about the equipment used. For instance, line 120 "a SUPRA 40 SEM" equipped with...? Configuration used?

line 123

I believe you meant "Bragg-Brentano" instead of "step-scanning mode"?

Concerning the explanation added (page 4, lines 127 - 133) on the texture. I understand that if the grains are large the rotation will not decrease the texture influence enough to have good statistics. However, even if the crystallite sizes are small, the texture can be strong enough to hinder an accurate analysis. This is true every time the intensities are used because the intensities are modified by the texture. The authors say that they used certain peaks of beta and alpha phases to reduce the effect of texture. However, one of the peaks used is alpha (100), which is one of the peaks one can see the intensity varying with the aging time. One only needs to check the 45 min, 1 h, 2 h, 4 h to see how much the intensity changes. It means that these intensities are not free of texture influence - or there is something else modifying the intensities. Moreover, if the intensities are not good for Rietveld refinement because of the texture, why would they not be considered influenced by texture for the analysis performed here?

Either I am missing something here or this explanation is not enough to justify using the intensities to "accurately" analyze the phases' fraction.

The authors answered that the XRD data was fitted. However, there is no fitting information, no fitted curves, no agreement factors of the fitting, in the manuscript.

line 169

It would be good to have TEM images to verify if omega particles are really not present.

SEM images (Figures 5, 6, and 7) should be improved.

Please, reformulate "It is apparent that there are very different morphological alpha precipitate ... is also evident".

The phrase is grammatically incorrect and it does not add any important information. Which are the different morphologies? What is the function of the alpha percentage on the aging time and temperature

Authors claim that "On the other hand, this raise is by an increase in the quantity of alpha-phase nuclei. On the other hand, this raise is by the growth of the existing alpha precipitate."

I am not sure what one should understand from this statement.

"It is possible that the distance between the particles decreases gradually with the value of alpha precipitate fraction increased."

It should be relatively easy to verify this by TEM images. The phrase is grammatically incorrect.

Where did equation 4 come from? A "bold and novel attempt" also needs an explanation of the logic of the idea.

line 302 - 303

How do the authors know that the "phase transition in the first stage is related to homogeneous nucleation"?

Concerning the hardening of the material, the authors explain that the increase of the hardening is due to "a kind of precipitation hardening generated by a great quantity of fine alpha phase precipitated in the beta matrix." How do the alpha precipitates increase the hardening of the material?

Did the authors make any measurement to show that the "alpha phase tends to become thicker and longer with further increasing aging time"? Saying that it has been observed before does not mean that this is what happens here.

Author Response

Response to the Reviewer’s Comments
We would like to thank the Reviewer for his/her helpful comments. We have followed all the suggestions and made revisions accordingly. The list below gives the details of the changes that we have made in the revised manuscript.

Point 1: The introduction contains a mixture of "beta-titanium", "beta-Ti", "Ti" alloys. Please, double-check if those alloys are really beta or just Ti.
Response 1: It has been described uniformly in the manuscript.

Point 2: Would you please add the unit to 10.3?
Response 2: According to the experimental formula [1]:
Mo Eq=1.0(wt.% Mo)+0.67 (wt.%V)+0.44 (wt.% W)+0.28 (wt.% Nb)+ 0.22 (wt.% Ta)+ 2.9 (wt.% Fe)+ 1.6(wt.% C)-1.0 (wt.% Al)
Therefore, there is no unit for the molybdenum equivalent of an alloy.
[1] Bania P.J. Beta titanium alloys and their role in the titanium industry. JOM. 1994, 46, 16-19.

Point 3: "being different from traditional metastable beta-Ti alloys. "How different? Different in which aspects?
Response 3: On the one hand, the molybdenum equivalence is often considered to evaluate the overall β stability for titanium alloy with various alloying element additions. In order to retain 100% β phase upon quenching from above the β transus, the molybdenum equivalence should be greater than 10. On the other hand, the molybdenum equivalent of the majority of commercial metastable β-titanium alloys is relatively high, which results in an increase in alloys cost, difficulty in smelting, inhomogeneous precipitation and slow aging response. The nominal molybdenum equivalence in the Ti–3.5Al–5Mo–4V alloy is only 10.3 and the total content of the β stabilizing elements is about 12%, which are almost the lowest values in the traditional metastable β-titanium alloys.

Point 4: table 1 Why different elements have different accuracy if, I suppose, they were measured under the same conditions in the same equipment?
Response 4: For the different alloy elements, the standards of their detection methods are different, and the testing instruments are also different. For example, the detection of carbon and sulfur was based on the GB/T 4698.14-2011, and the infrared carbon-sulfur analyzer was used as the test instrument; the detection of molybdenum content according to the GB/T 4698.19-2017 by spectrophotometer; and the chromium element was detected by burette according to the GB/T 4698.10-1996. On the other hand, the content of impurity elements is low, and it has a negative effect on the properties of the alloy, so its precision is higher.

Point 5: Please, add more technical information about the equipment used. For instance, line 120 "a SUPRA 40 SEM" equipped with...? Configuration used?
Response 5: According to the reviewer’s suggestion, we have made a further supplement:
The microstructures of the solution–treated samples were illustrated with an OLYMPUS Gx51 optical microscope (OM).
The microstructures of the aging–treated samples were characterized using a ZEISS SUPRA40 field emission scanning electron microscope (FESEM) at 10 kV under InLens mode with a protective pure nitrogen atmosphere at room temperature.

Point 6: I believe you meant "Bragg-Brentano" instead of "step-scanning mode"?
Response 6: We have made corrections:
The XRD was performed in Bragg-Brentano mode with a step size of 2°/min and covering 2θ values in the range of 25–80° on a Bruker D8 Advance by Cu Kα radiation (λ=1.5406 Å) at 40 kV and 40 mA at room temperature.

Point 7: Concerning the explanation added (page 4, lines 127 - 133) on the texture. I understand that if the grains are large the rotation will not decrease the texture influence enough to have good statistics. However, even if the crystallite sizes are small, the texture can be strong enough to hinder an accurate analysis. This is true every time the intensities are used because the intensities are modified by the texture. The authors say that they used certain peaks of beta and alpha phases to reduce the effect of texture. However, one of the peaks used is alpha (100), which is one of the peaks one can see the intensity varying with the aging time. One only needs to check the 45 min, 1 h, 2 h, 4 h to see how much the intensity changes. It means that these intensities are not free of texture influence - or there is something else modifying the intensities. Moreover, if the intensities are not good for Rietveld refinement because of the texture, why would they not be considered influenced by texture for the analysis performed here?
Either I am missing something here or this explanation is not enough to justify using the intensities to "accurately" analyze the phases' fraction.
Response 7: The strong peaks of the α phase appear at about 35°, 38°, 40°, 53°, 63°, 71°, 74°, 76°, and 77°. The strong peaks of the β phase appear at about 39°, 56°, and 70°. If there is no texture, the intensity of the peak should increase with the increase of aging time. However, due to the existence of texture, the position of the strong peak is not fixed, so the volume fraction of α phase cannot be determined based on a single peak. In order to obtain the volume fraction of the α phase more accurately, multiple peaks should be selected for calculation rather than only one peak.

Point 8: The authors answered that the XRD data was fitted. However, there is no fitting information, no fitted curves, no agreement factors of the fitting, in the manuscript.
Response 8: According to the reviewer’s suggestion, typical profile fitting results of the XRD patterns of the Ti–3.5Al–5Mo–4V alloy aged at 500°C for different times after solution treatment were supplemented. Please see Figure 9 in the section of 3.3.

Point 9: It would be good to have TEM images to verify if omega particles are really not present.
Response 9: The strong peaks of the ω phase mainly appeared at about 39°, 57°, 66°, 71°, and 79°. It is seen from Figure 3 that the peak of the ω phase was not observed, indicating that there is only a single β phase for the Ti–3.5Al–5Mo–4V alloy after solution treatment.

Point 10: SEM images (Figures 5, 6, and 7) should be improved.
Response 10: SEM images (Figures 5, 6, and 7) had been improved.

Point 11: Please, reformulate "It is apparent that there are very different morphological alpha precipitate ... is also evident".
The phrase is grammatically incorrect and it does not add any important information. Which are the different morphologies? What is the function of the alpha percentage on the aging time and temperature?
Response 11: For this phrase, we have made some revisions, please seen the section of 3.3 in the manuscript.
It was apparent that the size and amount of the α phase developed in the β matrix is a function of the aging time for the Ti–3.5Al–5Mo–4V alloy solution-treated at 850 and 1050°C from Figures 7 and 8. In order to describe quantitatively the relationship between the volume fraction of the α phase and aging time, the value of the α precipitate fraction in the Ti–3.5Al–5Mo–4V alloy under different aging treatments was calculated with the direct comparison methodology by Equations (1) and (2).

Point 12: Authors claim that "On the one hand, this raise is by an increase in the quantity of alpha-phase nuclei. On the other hand, this raise is by the growth of the existing alpha precipitate."
I am not sure what one should understand from this statement.
"It is possible that the distance between the particles decreases gradually with the value of alpha precipitate fraction increased." It should be relatively easy to verify this by TEM images.
Response 12: According to the Figure 7, the α phase was only precipitated locally when the aging time is short (Figure 7(a) and (b)). With increasing the aging time, the size and quantity of α phase increased (Figure 8 and Figure 7(c) to (f)). The increase in the quantity means an increase in the number of nucleation, and the increase in the size means the growth of the existing α phase. At the same time, the distance between the α particles decreased gradually as the aging time increased.

Point 13: Where did equation 4 come from? A "bold and novel attempt" also needs an explanation of the logic of the idea.
Response 13: Equation 4 comes from a book [1].
[1].Sha W.; Malinov S. Titanium Alloys: Modelling of microstructure, properties and applications, Woodhead Publishing, Abington Hall, Abington, Granta Park, Great Abington, Cambridge CB21 6AH, UK, 2009; 140.

Point 14: How do the authors know that the "phase transition in the first stage is related to homogeneous nucleation"?
Response 14: According to a series of literature [1-5], the kinetic parameters of JMAK deduced different transformation mechanisms. Furthermore, we have made a further supplement according to the Figure 5 that the first stage belongs to mixed transformation mechanisms: homogeneously nucleated and sheet–like grown α structure, and grain boundaries nucleated and grown α precipitate.
[1] Angelier C.; Bein S., Béchet J. Building a continuous cooling transformation diagram of β-CEZ alloy by metallography and electrical resistivity measurements. Metall. Mater. Trans. A 1997, 28, 2467-2475.
[2] Malinov S.; Sha W.; Markovsky P. Experimental study and computer modelling of the β⇒α+β phase transformation in β21s alloy at isothermal conditions. J. Alloys Compd. 2003, 348, 110–118.
[3] Malinov S.; Guo Z.; Sha W.; Wilson A. Differential scanning calorimetry study and computer modeling of β⇒α phase transformation in a Ti-6Al-4V alloy. Metall. Mater. Trans. A 2001, 32, 879–887.
[4] Sha W.; Malinov S. Titanium Alloys: Modelling of microstructure, properties and applications, Woodhead Publishing, Abington Hall, Abington, Granta Park, Great Abington, Cambridge CB21 6AH, UK, 2009; 125–140.
[5] Naveen M.; Santhosh R.; Geetha M.; Rao M.N. Experimental study and computer modelling of the β→α+β phase transformation in Ti15-3 alloy under isothermal conditions. J. Alloys Compd. 2014, 616, 607-613.

Point 15: Concerning the hardening of the material, the authors explain that the increase of the hardening is due to "a kind of precipitation hardening generated by a great quantity of fine alpha phase precipitated in the beta matrix." How do the alpha precipitates increase the hardening of the material?
Response 15: On the one hand, the α phase is a solid solution with hexagonal closely packed crystal structure, which limited the number of operative slip systems. On the other hand, the great quantity of fine α phase precipitating in the β matrix provided more α/β interfaces, which could act as dislocation barriers and effectively hinder the dislocation movement. Therefore, the precipitation of α phase had a significant precipitation hardening effect [1-2].
[1] Tan Y.B.; Tian C.; Liu W.C.; Xiang S.; Zhao F.; Liang Y.L. Effect of Rolling Temperature on the Microstructure and Tensile Properties of 47Zr-45Ti-5Al-3V Alloy. J. Mater. Eng. Perform. 2018, 27, 1803-1811.
[2] Dong R.F.; Li J.S.; Kou H.C.; Fan J.K.; Tang B.; Sun M. Precipitation behavior of α phase during aging treatment in a β-quenched Ti-7333. Mater. Charact. 2018, 140, 275-280.

Point 16: Did the authors make any measurement to show that the "alpha phase tends to become thicker and longer with further increasing aging time"? Saying that it has been observed before does not mean that this is what happens here.
Response 16: According to the reviewer’s suggestion, we have made a further supplement, please seen the Figure 8 in the section of 3.2.
